# Salmonellosis Outbreak in a Rottweiler Kennel Associated with Raw Meat-Based Diets

**DOI:** 10.3390/ani15213196

**Published:** 2025-11-03

**Authors:** Betina Boneva-Marutsova, Plamen Marutsov, Marie-Louise Geisler, Georgi Zhelev

**Affiliations:** 1Department of Veterinary Microbiology, Infectious and Parasitic Diseases, Faculty of Veterinary Medicine, Trakia University, 6000 Stara Zagora, Bulgaria; plamen.marutsov@trakia-uni.bg (P.M.); g.zhelev.georgiev@trakia-uni.bg (G.Z.); 2Laboklin GmbH & CO.KG, 97688 Bad Kissingen, Germany; m.geisler@laboklin.com

**Keywords:** *Salmonella* Agona, dogs, raw meat-based diet (RMBD), gastroenteritis, antimicrobial susceptibility, zoonosis

## Abstract

This report outlines an outbreak of salmonellosis in a Rottweiler breeding kennel caused by *Salmonella* Agona, which was linked to an unlicensed raw meat-based diet. It emphasises the importance of safe feeding practices, good biosecurity measures, and traceable feed sources in kennels. Additionally, it raises awareness about the zoonotic risks associated with raw feeding. The diagnostic and treatment protocols described can serve as a model for managing similar cases under the One Health approach.

## 1. Introduction

*Salmonella* is a significant zoonotic pathogen known for its impact on both human and animal health. It comprises over 2600 different serovars, which are responsible for a range of diseases, including typhoidal and non-typhoidal salmonellosis [1,2]. This pathogen is primarily transmitted through the consumption of contaminated food or water, as well as through direct contact with infected animals or their environment [3]. In 2023, salmonellosis was identified as the second most prevalent bacterial zoonosis in humans, with a total of 77,486 confirmed cases across European Union Member States. *Salmonella* was the agent linked to the highest number of food-borne outbreaks, with a total of 9210 confirmed cases. Among these, 1726 cases (18.7%) resulted in hospitalisation, and there were 16 deaths, resulting in a case fatality rate of 0.17% [4].

Companion animals, particularly dogs, can act as reservoirs for *Salmonella* spp., posing a potential risk for human infection. Data from the European Food Safety Authority (EFSA) and European Centre for Disease Prevention and Control (ECDC) indicate that the prevalence in dogs is reported at 2.7% in EU Member States. In contrast, non-EU countries reported a higher prevalence of 4.3%, with 46 out of 1082 dogs testing positive [5].

*S. enterica* does not seem to have developed specific adaptations to canine hosts, as a diverse array of serovars has been detected in dogs. The distribution of these serovars varies by region and country, likely due to differences in dietary habits and environmental sources that influence exposure to various sources of the bacteria [6,7]. In dogs, sporadic cases of salmonellosis are primarily observed, with most infections being asymptomatic. While many cases remain subclinical, clinical signs can emerge as acute gastroenteritis, especially during stressful situations such as parturition, lactation, or dietary changes [8].

*S.* Agona (group B1) is a lesser-known serovar that has been linked to foodborne illness outbreaks in the EU. From 2007 to 2016, it was implicated in 13 outbreaks, which resulted in a total of 636 illnesses requiring hospitalisation. Of these, nine outbreaks were attributed to the consumption of contaminated foods [9]. The frequency of *S.* Agona isolation in dogs was found to be 13.31%, with 5 out of 442 *S. enterica* isolates in a study conducted in UK [6]. The serovar Agona has been implicated in multiple foodborne and waterborne outbreaks worldwide and is frequently associated with contaminated feed and animal-derived products [10,11].

The clinical signs observed in the dogs included depression, vomiting, and diarrhoea, sometimes with streaks of blood. Fever was not a common finding, and none of the dogs showed an elevated temperature. Because the dogs were infected and shed *Salmonella* in the environment, there was a risk of transmitting the infection to any humans with whom they came into close contact.

The Implications of Such Outbreaks Can Be Serious, Affecting Not Just the Health of the Dogs Involved but Also Raising Concerns About Salmonella Outbreaks Among Human Populations.

## 2. Case Description

### 2.1. Housing

The kennel houses a population of nine Rottweilers, comprising three males and six females, three of which had recently whelped. Each dog was housed individually in a cage measuring 2.5 × 3.2 m^2^. Regular cleaning procedures included daily waste removal and washing the floor with running water. Food and water bowls were thoroughly cleaned and disinfected. Additionally, the kennel was thoroughly cleaned once or twice a week with diluted bleach, depending on the level of dirt.

### 2.2. Feeding

The adult dogs are primarily fed a raw meat-based diet (RMBD) from unlicensed origin, typically composed of a mixture of beef, chicken, turkey, and offal collected from various slaughterhouses. The meat is stored frozen in pre-portioned packages. Once thawed, the portions are kept refrigerated and used within three to four days.

### 2.3. Veterinary Care, Prophylaxis and Supervision

Preventive health management within the kennel focuses on routine deworming and the administration of mandatory core vaccinations in accordance with standard veterinary protocols.

### 2.4. History

#### 2.4.1. Litter A

The epidemiological narrative of the outbreak began with the parturition of a five-year-old female dog. According to the owner’s recollections, approximately one-week post-partum, the animal exhibited clinical signs of gastrointestinal disease. These signs included lethargy, vomiting, and diarrhoea, with faeces described as mushy or watery. Despite maintaining a normal appetite, the bitch experienced weight loss, and insufficient milk production necessitated supplementing the puppies with a milk replacer. The attending veterinarian prescribed metronidazole at a dosage of 10–15 mg/kg administered orally every 12 h, which led to resolution of the dam’s clinical signs. However, around 48 h following the onset of the dam’s illness, clinical signs of vomiting and diarrhoea emerged in the puppies, with all eight individuals in the litter affected within 36 h. The local veterinarian initially treated them with metronidazole, which only provided temporary improvement; clinical signs rapidly reappeared, with increased vomiting frequency and progressively watery, bloody faeces. Although the puppies continued to show appetite, their growth rates appeared to be delayed. One puppy exhibited more severe clinical signs, including lethargy, reduced appetite, and signs of marked dehydration. The local veterinarian observed pale mucous membranes, prolonged capillary refill time, and haematochezia. To further investigate, he conducted a rapid diagnostic test for canine parvovirus, coronavirus, and *Giardia duodenalis*. The test results returned positive for *Giardia*. Consequently, a therapeutic regimen comprising fenbendazole (50 mg/kg SID for 3 days), sulphonamide (15 mg/kg bodyweight BID for 5 days), and metronidazole (25 mg/kg BID, for 5 days) was instituted. Additionally, to restore the blood’s fluid volume, electrolyte solutions were administered subcutaneously and orally.

This treatment led to an improvement in the affected puppies; however, complete resolution of clinical signs was not achieved.

#### 2.4.2. Litter B

A comparable situation occurred in litter B 16 days later. In this case, a two-year-old dam successfully delivered seven neonates. Shortly after parturition, the dam developed clinical signs of illness, which subsequently manifested in the offspring. The clinical course in the neonates was notably more severe, with all puppies exhibiting overt signs of disease. The earlier diagnosis of giardiasis in litter A served as the basis for administering fenbendazole and metronidazole to these animals as well. This treatment resulted in temporary clinical improvement lasting approximately 24–48 h, after which a recurrence of symptoms was observed. The local veterinarian repeated the rapid diagnostic tests and performed more in-depth parasitological examinations, all of which yielded negative results. These findings indicated that the observed clinical presentation was unlikely to be due to a recurrence of the previous infection.

#### 2.4.3. Litter C

Since clinical symptoms have only been observed in the dogs that gave birth and their offspring, to minimise potential risks to the last pregnant dog, the owner moved her away from the others. Three weeks before parturition, the dam had been isolated in a designated maternity ward as a preventive measure to mitigate potential health risks; however, this approach was ultimately unsuccessful. The puppies from litter C, along with their three-year-old dam, were the last to show clinical signs of health issues. In this third litter, which consists of ten puppies, the owner has noted a recurrence of the same health problems observed previously.

Meanwhile, gastrointestinal symptoms were observed in the remaining six adult dogs in the kennel, comprising three males and three non-pregnant females. These dogs were affected within a period of 27 days between the birth of the second and third female. The owner, however, is unable to recall the precise order in which each of them became ill. The clinical symptoms exhibited by these dogs were comparable, although they were generally milder than those that had been observed in previous cases. Among them, the most severely affected was a 15-month-old female (Cara) which exhibited vomiting, bloody diarrhoea, polyuria/polydipsia, anorexia, reduced activity levels, and a hunched posture. Cara received additional treatment with amoxicillin in standard doses, which resulted in improvement in her condition.

As the health situation remained unresolved and in view of the conflicting diagnostic findings, the owner decided to seek additional veterinary assistance. Subsequently, a team of veterinarians from the Department of Veterinary Microbiology, Infectious and Parasitic Diseases at Trakia University conducted an on-site inspection of the kennel and performed further investigations.

### 2.5. Physical Examination

During the general examination, a range of clinical signs were observed in the puppies from litter C, as well as in their mother. The neonate puppies exhibited faecal staining due to diarrhoea, some of them presenting with abdominal distension, which later progressed to sunken bellies. Frequent vocalisations accompanied by whimpering and decreased appetite were observed. None of the puppies in the litter or the bitch showed signs of hyperthermia. Upon abdominal palpation, the puppies displayed abdominal guarding, indicative of visceral pain. These clinical findings supported the need for further diagnostic investigation to determine the underlying cause of the gastrointestinal disorder.

### 2.6. Diagnostics Performed

During the field visit, individual samples were collected from all adult dogs and six neonatal puppies. Each sample set consisted of approximately 10 g of faeces, accompanied by faecal swabs and vomitus, for comprehensive bacteriological examination. To aid in source tracing of the outbreak, environmental and food samples were also obtained. These included portions of raw food (*N* = 3 pooled samples), as well as swabs from food bowls (*N* = 6), floors (*N* = 6), and cage walls (*N* = 6). All samples were initially cultured in both liquid enrichment media (Tryptic Soy Broth and Selenite Broth) and solid media (Blood Agar Base and MacConkey agar, HiMedia™, Maharashtra, India). Incubation was conducted under both aerobic and anaerobic conditions at 37 °C for 24 h to encompass a broad spectrum of bacterial growth Lactose-negative colonies grown on MacConkey agar were specifically selected for further examination as potential *Salmonella* isolates. Serological typing of suspected *Salmonella* colonies by slide agglutination revealed that they belonged to group B. In partnership with Laboklin in Germany, the samples were sent for additional analysis. The procedure begins with transferring 10 mL of the broth and vortexing. Then they are streaked out onto an XLD agar plate (Thermo Scientific™ XLD Agar, Vantaa, Finland) and are performing liquid enrichment using Tetrathionate and Selenite broth. Incubation occurs at 42 °C for 24 h. Additionally, 3 mL of the mixed bouillon is transferred into Tetrathionate and Selenite bouillon for liquid enrichment. After this, a loopful of the enriched broth was then subcultured onto both XLD agar and CHROMagar Salmonella (Becton Dickinson GmbH, Heidelberg, Germany), with an additional incubation for 24 h at 42 °C. Post-incubation, CHROMagar plates were inspected for violet colonies, whereas XLD plates were examined for black colonies characteristic of *Salmonella* spp. Suspect colonies were further subcultured on Hektoen agar (Thermo Scientific™ Hektoen, Vantaa, Finland) and CHROMagar. Genus-level identification was performed using MALDI-TOF mass spectrometry (MALDI Biotyper^®^ sirius one; Bruker). Serological typing of confirmed *Salmonella* isolates was conducted according to the classical White–Kauffmann–Le Minor scheme [12], using slide agglutination with specific O- and H-antigen sera (Sifin Diagnostics, Berlin, Germany).

## 3. Results

*Salmonella enterica* serovar Agona was isolated from all types of collected specimens, including faeces, vomitus, raw food samples, and environmental swabs. Notably, particularly high bacterial loads were detected in both frozen and thawed samples of biologically appropriate raw food (BARF), indicating that the contaminated feed was a likely source and reservoir of the pathogen within the kennel environment.

Antimicrobial susceptibility testing of the isolate was performed using the broth microdilution method in accordance with CLSI guidelines (Table 1). The *Salmonella enterica* serovar Agona isolate demonstrated susceptibility to the majority of tested antimicrobials, with notable resistance to cephalexin, most aminoglycosides, macrolides, lincosamides, and fusidic acid, and intermediate susceptibility to polymyxin B.

Additional diagnostic tests for common gastrointestinal and infectious diseases, including canine parvovirus, canine coronavirus, giardia, and helminths, yielded negative results.

### 3.1. Human Cases

During the epidemiological investigation, an employee reported past clinical signs consistent with gastroenteritis with nausea, chills, and diarrhoea. These symptoms lasted approximately five days, and more than two weeks have since passed. The employee was referred to the Regional Health Inspectorate, but tests showed a negative result for *Salmonella* spp.

### 3.2. Outcome

Treatment and Control Measures:

Feeding of raw food products was immediately discontinued due to the confirmed *Salmonella* contamination, and all affected batches were sent for destruction. Antimicrobial therapy was initiated for all dogs presenting with diarrhoea and systemic signs, with treatment regimens tailored according to the antimicrobial susceptibility profile of the isolate. Puppies received oral amoxicillin–clavulanic acid suspension at a dosage of 12.5 mg/kg twice daily, while adult dogs were treated with oral enrofloxacin at 5 mg/kg once daily. The treatment duration ranged from 5 to 7 days.

All kennel personnel and owners underwent screening for *Salmonella* spp., coordinated by the Regional Health Inspectorate, and all results were negative. As part of the outbreak control strategy, enhanced environmental hygiene and disinfection protocols were implemented. These included the temporary removal of all dogs, preliminary disinfection procedures, followed by thorough cleaning and final surface disinfection. The kennel environment was subjected to several rounds of treatment using a commercial broad-spectrum disinfectant formulation containing glutaraldehyde and a blend of quaternary ammonium compounds, applied at manufacturer-recommended concentrations. Following the completion of treatment, all dogs were tested twice for *Salmonella* carriage at 10-day intervals. The results of both follow-up tests were negative, confirming the successful elimination of the pathogen from the kennel population.

## 4. Discussion

Salmonellosis is not commonly recognised as a cause of enteric disorders in dogs [13,14]. Research shows that *Salmonella* spp., bacterial carriage is relatively low in healthy dogs fed commercial kibble or canned food [15]. However, there is an increasing trend among pet owners who opt to feed their dogs raw meat. This practice raises important questions and concerns about the potential risks associated with raw diets, particularly regarding the contamination and spread of pathogenic bacteria such as *Salmonella*, *Campylobacter*, and *Listeria* to humans through contact with contaminated food, pets or their faeces [11,14,15]. The current case confirms that feeding RMBD to dogs, especially those of unknown origin, can significantly increase the risk of *Salmonella* infection. This not only poses serious health issues for pets but also leads to ongoing faecal shedding and represents a significant risk to human health. *Salmonella* Agona is rarely reported in dogs according to the scientific literature. In contrast, other serovars, such as Typhimurium, Dublin, Enteritidis, and Montevideo, are more commonly isolated from canine cases [6]. However, *S.* Agona has been associated with multiple foodborne outbreaks in humans and is recognised as the serovar requiring the lowest infectious dose to cause disease following ingestion of contaminated food [16,17]. While this has not yet been demonstrated in dogs, it is possible to speculate on its implications. S Agona has emerged as the 13th of 20 most frequently reported serovars in human salmonellosis within the EU between 2021 and 2023, underscoring a critical public health challenge that demands urgent attention [4].

More than 35 days passed between the index case and the last bitch and her litter being affected. Hypothetically, all dogs were exposed to *Salmonella* via the diet initially, with clinical disease appearing in the newly parturient bitches and their pups. This may suggest that *S*. Agona is not well adapted and is not obligately pathogenic for dogs. The temporal association between parturition and onset of illness suggests that physiological stress, immunosuppression, and environmental changes may have contributed to disease expression, as previously proposed for other outbreaks [3]. The occurrence of disease in the remaining adult dogs, including males and non-pregnant females, was most likely associated with prolonged dietary exposure to *Salmonella* Agona, increased infectious pressure from affected animals, environmental contamination, or the potential development of partial strain adaptation to the canine host over time. *Salmonella* spp. prefers to grow at a pH above 5.5 [18] and can be killed at a pH below 3 [19]. Gastric pH in dogs is reported to be 3.0 in the prefeeding, and 1.8 in the postfeeding period [20], which can inhibit the survival of potential pathogens such as *Salmonella* and impact infection and the development of clinical illness.

The clinical signs observed—vomiting, diarrhoea, weight loss, pale mucous membranes, polyuria, polydipsia, weakness, lethargy, and dehydration—align with findings reported by other researchers [6,21]. Some researchers have noted a rapid progression of symptoms characterised by a sudden onset of apathy, loss of appetite, and bloody diarrhoea. This serious condition can lead to death within just 12 to 36 h after the initial symptoms appear. *Salmonella* spp., isolates obtained in these cases were serotyped as *S.* Typhimurium, *S.* Infantis, and *S.* Heidelberg [14]. *S*. Agona was isolated from faeces, vomitus, food, and multiple environmental samples, confirming a persistent contamination of both the feed and kennel environment. The finding of high bacterial loads in both frozen and thawed RMBD samples underscores the risk associated with inadequate sourcing, storage, and handling of animal-derived feeds. Comparable feed-borne kennel outbreaks have been previously described [8], further emphasising the importance of feed traceability and supplier certification. In this case, the raw food was a mix that included turkey meat, where serovar Agona was identified as the most prevalent isolate, accounting for 48.7% of cases. In contrast, this strain was found to rank significantly lower in other animal species [4].

In the aforementioned outbreak, all affected dogs achieved full clinical recovery, and none became carriers of the pathogen. Following the implementation of control measures all tested dogs were negative for *Salmonella* spp. A comparable outcome was confirmed in our case. Although salmonellosis is generally a self-limiting disease, antimicrobial treatment is recommended in cases of severe or invasive infection [22,23]. In contrast, under experimental conditions, dogs infected with *Salmonella* may shed the bacteria for a longer period—up to 24 days [24].

Research on canine salmonellosis primarily highlights the association between infection and the consumption of raw meat-based diets (RMBDs). However, certain dog treats may also pose comparable health risks. A study conducted in the United States detected *Salmonella* spp., in 4 out of 505 tested pet food samples (0.79%), with contamination found in pig ear treats originating from South America. Among these isolates, only one was identified as *S. enterica* serovar Agona [25]. These findings emphasise the importance of preventive strategies aimed at reducing the risk of infection. Such measures should include educating pet owners about the potential hazards of feeding raw meat, encouraging the use of safer commercial diets, and enforcing proper food handling and hygiene practices to minimise exposure to contaminated materials. Although numerous reports highlight multidrug-resistant (MDR) *Salmonella* [3,26], the *S.* Agona isolate recovered in this study exhibited susceptibility to most tested antimicrobials. Nevertheless, resistance to cephalexin, several aminoglycosides, macrolides, lincosamides, and fusidic acid, together with intermediate susceptibility to polymyxin B, remains a cause for concern. These findings emphasise the importance of prudent antimicrobial stewardship and laboratory-guided therapy to prevent the emergence and spread of resistant strains.

After discontinuation of raw feeding, implementation of disinfection protocols, and administration of targeted antimicrobial therapy, all dogs achieved full clinical recovery and repeatedly tested negative for *Salmonella* carriage. These outcomes support that early intervention, feed control, and hygiene reinforcement are effective containment measures. Nonetheless, the potential for asymptomatic carriage and environmental persistence of *Salmonella* warrants routine post-outbreak monitoring [24].

Although all human contacts tested negative, *Salmonella* transmission from dogs to humans is well documented [24]. Kennel workers and owners exposed to contaminated feed, faeces, or fomites—as confirmed in our case—remain at risk, emphasising the need for strict hygiene education, personal protective practices, and regular environmental disinfection [27].

### Epidemiological and One Health Implications

This outbreak exemplifies the One Health relevance of raw feeding practices. The detection of *S.* Agona in both animal and environmental samples underlines the interconnectedness of food safety, animal health, and public health. Preventive measures should include:Certification and traceability of all feed and raw meat sources.Routine microbiological monitoring of feed, kennel environments, and faecal samples.Education of pet owners and staff regarding safe food handling and zoonotic risk.Biosecurity reinforcement during high-risk periods such as parturition or lactation.Prudent antibiotic use aligned with antimicrobial stewardship principles.

## 5. Conclusions

This case emphasises the epidemiological importance of feed safety and biosecurity in kennels and underscores the zoonotic potential of *S.* Agona. Comprehensive diagnostic work-up, including MALDI-TOF-MS identification and susceptibility testing, allowed for precise therapeutic intervention and successful outbreak resolution. The findings reinforce the need for continuous surveillance of raw pet foods and public awareness of their risks within the broader One Health framework.

## Figures and Tables

**Table 1 animals-15-03196-t001:** Antimicrobial susceptibility profile of the *Salmonella enterica* serovar Agona isolate obtained from the kennel outbreak (broth microdilution method, CLSI).

Antimicrobial Agent	MIC (µg/mL)	Interpretation	Antimicrobial Agent	MIC (µg/mL)	Interpretation
Penicillin G	0	R	Difloxacin	≤0.5	S
Ampicillin	2	S	Enrofloxacin	≤0.5	S
Amoxicillin	n/a	S	Marbofloxacin	≤1	S
Amoxicillin + Clavulanic acid	2/1	S	Ofloxacin	n/a	S
Cephalexin	>4	R	Pradofloxacin	≤0.25	S
Cefovecin	2	S	Orbifloxacin	≤1	S
Gentamicin	≤2	S	Tetracycline	n/a	S
Neomycin (Framycin)	>8	R	Doxycycline	4	S
Kanamycin	0	R	Florfenicol	n/a	S

Legend: S = sensitive; R = resistant Testing according to CLSI; The interpretation of the results of the antibiotic sensitivity test is based on clinical breakpoints for systemic use.

## Data Availability

The original contributions presented in this case are included in the article. Further inquiries can be directed to the corresponding author.

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
