# Peer review of "Salmonellosis Outbreak in a Rottweiler Kennel Associated with Raw Meat-Based Diets"

_animals, 2025, doi:10.3390/ani15213196_

Round 1
Reviewer 1 Report
Comments and Suggestions for Authors
The paper is a case report describing an outbreak of salmonellosis in a breeding dog kennel, caused by contaminated raw food. The case is of practical interest to veterinarians and practitioners. The description of the outbreak, clinical investigation, diagnosis, and subsequent treatment is well written and comprehensive. The topic is timely and contributes to the ongoing discussion on the advantages and risks of raw food diets in dogs. I would recommend only minor revisions before considering the paper for publication.
Line 21: change to vomit
Line 22: were collected and examined
Line 30: some instead of most?
Line 48, 64, 75 and so on: please separate citation numbers by commas not semicolons
Line 77-78: please rewrite the sentence to provide more detailed information on the clinical presentation (briefly describe the symptoms), and then address the carrier state and its consequences.
Line 85: what were the good hygienic conditions – please add short info and then check with lines 228-230
Line 87: please clarify “unlicensed origin” — does it refer to meat from butcheries or animals that died on farms?
Lines 91 – 93: Please present that information in a separate paragraph (veterinary care/prophylaxis/ supervision).
Line 106: treated …?puppies?
Line 109: appeared to be delayed
Lines 125-128: sentence is repeated
Author Response
Dear Reviewers,
We sincerely thank you for your evaluation of our manuscript and for your constructive comments. We appreciate your commitment and time, as well as the opportunity to further improve the clarity and completeness of our study. The attached file presents all the changes made in accordance with your comments, highlighted.
Kind Regards!
Line 21: change to vomit - Corrected
Line 22: were collected and examined - Corrected
Line 30: some instead of most? –corrected in the attached file
Line 48, 64, 75, and so on: please separate citation numbers by commas, not semicolons - corrected
Line 77-78: please rewrite the sentence to provide more detailed information on the clinical presentation (briefly describe the symptoms), and then address the carrier state and its consequences.- corrected in attached file
Line 85: what were the good hygienic conditions – please add short info and then check with lines 228-230 - corrected
Line 87: please clarify “unlicensed origin” — does it refer to meat from butcheries or animals that died on farms? - corrected
Lines 91 – 93: Please present that information in a separate paragraph (veterinary care/prophylaxis/ supervision).- corrected
Line 106: treated …?puppies? - corrected
Line 109: appeared to be delayed - corrected
Lines 125-128: sentence is repeated - corrected

Reviewer 2 Report
Comments and Suggestions for Authors
The manuscript is well-written, clear, and objective. Its impact lies in its status as the first report of a salmonellosis outbreak in kenneled dogs associated with raw meat-based diets and the "one health" approach. Its relevance remains high since Salmonella Agona has been identified as an emerging serotype causing salmonellosis outbreaks in Europe, including one in July 2025.
However, some information needs to be included to make the manuscript clearer to the reader.
How many puppies were affected per litter (litters A and B)? All of them?
In history, I suggest presenting the chronological order of events. It was unclear how long after the first case adult males and non-pregnant females showed clinical signs. This is important, as the discussion mentions, "The occurrence of disease in the remaining adult dogs, including males and non-pregnant females, was most likely associated with prolonged dietary exposure to Salmonella Agona..."
Were there differences in the clinical signs presented by adult males, non-pregnant females, whelped females, and puppies? If so, I suggest presenting this differentiation in a table. This is optional, only if there are differences in clinical signs between animal categories.
The table should be improved by separating the antimicrobial agent, MIC (µg/mL), and interpretation into different columns.
Line 248: Please remove "etc."
In discussion, it is important to emphasize that S. Agona is among the 10 most isolated serotypes in food-animal source since 2022 in Europe, according to reference 4.
Author Response
Dear Reviewers,
We sincerely thank you for your evaluation of our manuscript and for your constructive comments. We appreciate your commitment and time, as well as the opportunity to further improve the clarity and completeness of our study. The attached file presents all the changes made in accordance with your comments, highlighted.
Kind Regards!
However, some information needs to be included to make the manuscript clearer to the reader.
How many puppies were affected per litter (litters A and B)? All of them?- corrected in the attached file
/For litter A “with all eight individuals in the litter affected within 36 hours”
For litter B “all puppies exhibiting overt signs of disease”/
In history, I suggest presenting the chronological order of events. It was unclear how long after the first case adult males and non-pregnant females showed clinical signs. This is important, as the discussion mentions, "The occurrence of disease in the remaining adult dogs, including males and non-pregnant females, was most likely associated with prolonged dietary exposure to Salmonella Agona..." - corrected
Were there differences in the clinical signs presented by adult males, non-pregnant females, whelped females, and puppies? If so, I suggest presenting this differentiation in a table. This is optional, only if there are differences in clinical signs between animal categories. – There are no significant differences in clinical signs.
The table should be improved by separating the antimicrobial agent, MIC (µg/mL), and interpretation into different columns. - corrected
Line 248: Please remove "etc." - corrected
In discussion, it is important to emphasize that S. Agona is among the 10 most isolated serotypes in food-animal source since 2022 in Europe, according to reference 4. - corrected
